# rNCA: Self-Repairing Segmentation Masks

**Malte Silbernagel**[*1]                                                                    MALTE.SILBERNAGEL@DI.KU.DK
**Albert Alonso**[*1]                                                                                        AAF@DI.KU.DK
**Jens Petersen**[1,2]                                                                                      PHUP@DI.KU.DK
**Bulat Ibragimov**[1]                                                                                    BULAT@DI.KU.DK
**Marleen de Bruijne**[1,3]                                                                           MARLEEN@DI.KU.DK
**Madeleine K. Wyburd**[1]                                                                            MAWY@DI.KU.DK

[1] *Department of Computer Science, University of Copenhagen, Copenhagen, Denmark*

[2] *Department of Oncology, Rigshospitalet, Copenhagen, Denmark*

[3] *Department of Radiology and Nuclear Medicine, Erasmus MC - University Medical Center Rotterdam, The Netherlands*

**Editors:** Accepted for publication at MIDL 2026

## Abstract

Accurately predicting topologically correct masks remains a difficult task for general segmentation models, which often produce fragmented or disconnected outputs. Fixing these artifacts typically requires handcrafted refinement rules or architectures specialized to a particular task. Here, we show that Neural Cellular Automata (NCA) can be directly repurposed as an effective refinement mechanism, using local, iterative updates guided by image context to repair segmentation masks. By training on imperfect masks and ground truths, the automaton learns the structural properties of the target shape while relying solely on local information. When applied to coarse, globally predicted masks, the learned dynamics progressively reconnect broken regions, prune loose fragments and converge towards stable, topologically consistent results. We show how refinement NCA (rNCA) can be easily applied to repair common topological errors produced by different base segmentation models and tasks: for fragmented retinal vessels, it yields 2–3% gains in Dice/clDice and improves Betti Errors, reducing $\beta_0$ errors by 60% and $\beta_1$ by 20%; for myocardium, it repairs 61.5% of broken cases in a zero-shot setting while lowering ASSD and HD by 19% and 16%, respectively. This showcases NCA as effective and broadly applicable refiners.

**Keywords:** neural cellular automata, segmentation refinement, topology preservation

---

* Contributed equally

. Code available at www.github.com/maltesilber/rnca

## 1. Introduction

Modern segmentation networks achieve strong pixel- and voxel-level accuracy across medical tasks, but often fail to preserve topological correctness when dealing with thin branches, closures, and continuity (Long et al., 2015; Ronneberger et al., 2015; Huang et al., 2019; Yuan et al., 2020a; Liu et al., 2021). This results in fragmented masks with gaps degrading their reliability for quantitative downstream image processing tasks, critical for many clinical applications (Wyburd et al., 2024).

To mitigate such failures, many works modify the segmentation model and the training pipelines themselves through, for example, topology-aware losses, diffeomorphic transformations or surrogate representations or by focusing the architecture in a specific anatomical prior (Clough et al., 2020; Shit et al., 2021; Kervadec et al., 2021; Wyburd et al., 2021; Alonso and Kirkegaard, 2023). Of particular interest to the medical imaging community is the detection of thin tubular structures, where anatomically plausible solutions are paramount and challenging to enforce in generic models (Kirchhoff et al., 2024; Huang et al., 2024; Song et al., 2025; Amiri et al., 2024). While specialized methods improve structural fidelity, their strong ties to a particular problem limit their applicability across domains and make them harder to translate to new settings.

Because of this, many approaches leave the general segmentation model unchanged and use post-processing steps to improve output coherence. Classical post-processing methods promote local smoothness but struggle to repair complex topology (Boykov and Jolly, 2001; Soille et al., 1999), while learned refiners, such as CRF-RNN (Zheng et al., 2015), integrate similar operations into an end-to-end framework, yet remain limited to local consistency rather than topological repair. More recent work embeds topological priors directly into the refinement step (Zdyb et al., 2025; Dima et al., 2025), although these methods remain tailored to specific problem settings. In contrast to these, we treat mask repair as a problem-agnostic *learned local dynamical process* applied after segmentation. Neural Cellular Automata (NCA) (Mordvintsev et al., 2020) provide a natural framework for learning local update rules that iteratively evolve spatial states, and whose dynamics can be externally conditioned (Sudhakaran et al., 2022). Prior work has already shown that NCA can generate coherent end-to-end segmentations in both 2D and 3D (Kalkhof et al., 2023; Kalkhof and Mukhopadhyay, 2023; Ranem et al., 2025) for low compute solutions but often at the cost of reduced accuracy. Here, we use NCA as a plug-in refiner, referred to as rNCA, that learns iterative growth and pruning dynamics to repair initial segmentation masks. The method operates without retraining the base model and generalizes across seg-

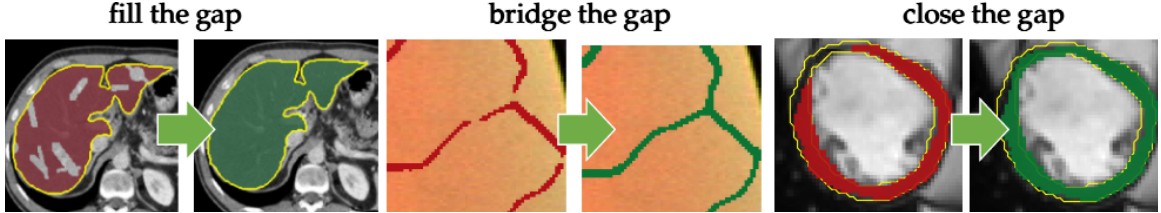

Figure 1: **Examples of topologies repairs achieved by rNCA**, including filling, reconnection and closing.

mentation architectures, yielding more connected and more structurally and anatomically plausible masks (Figure 1).

## 2. Related Work

**Neural Cellular Automata** NCA extend classical Cellular Automata by making the local update rule learnable (Neumann and Burks, 1966; Mordvintsev et al., 2020), enabling stable, self-organizing structures through iterative local updates. Most NCA work focuses on pattern growth, external conditioning, or spatio-temporal behaviors (Pajouheshgar et al., 2024; Sudhakaran et al., 2022; Pajouheshgar et al., 2023), typically for a single target shape or synthetic tasks. Recent extensions explore probabilistic variants (Palm et al., 2022; Faldor and Cully, 2024; Kalkhof et al., 2025) and lightweight biomedical applications (Kalkhof et al., 2023; Kalkhof and Mukhopadhyay, 2023; Ranem et al., 2024), though adoption in medical imaging remains limited. To the best of our knowledge, this is the first work to propose NCA as a refinement tool, potentially opening new directions for future research.

**Segmentation Refinement** Refinement has emerged as a practical alternative to modifying or retraining large segmentation models, allowing targeted correction of their systematic errors. Classical post-processing methods such as morphological filling and pruning (Soille et al., 1999) address only simple artifacts, whereas model-tied refiners (e.g., PointRend (Kirillov et al., 2020), RefineMask (Zhang et al., 2021), Deep Closing (Wu et al., 2024)) attach task-specific heads which require retraining. On the other hand, model-agnostic approaches act on noisy masks using boundary or global information (Tang et al., 2021; Yuan et al., 2020b; Zhou et al., 2020; Cheng et al., 2020; Wang et al., 2023) to further finetune segmentation in a single forward pass. An interesting perspective is that of iterative strategies (Lagergren et al., 2020, 2023), which show that repeated local updates can successfully stabilize topology and promote solutions that promote mask connectivity. Our work builds on the view that segmentation refinement is essentially the problem of learning local repair dynamics. NCA already operates through learned local updates that evolve a state toward stable shapes, which aligns directly with how topological corrections must be applied.

## 3. Method

We model segmentation refinement as an iterative local state evolution inspired by the NCA (Mordvintsev et al., 2020). Given an input image $x$ and an initial mask $\hat{y}_0$, the refiner maintains a per-pixel state vector $s_t$ with $K$ channels. The initial mask $\hat{y}_0$ may originate from an external segmentation model or from a perturbed version of the ground-truth labels, allowing the refiner to operate independently of the base segmentation method. The first channel of $s_t$ encodes the observable mask $\hat{y}_t$ and the remaining $K-1$ channels capture the internal dynamics of the system of timestep $t$. An overview is shown in Figure 2.

### 3.1. Design philosophy

The model does not encode any task-specific topology. Instead, it learns generic repair dynamics based on the perturbative states seen during trainings. This keeps the refiner architecture-agnostic, dataset-agnostic and simple to integrate into existing pipelines. We

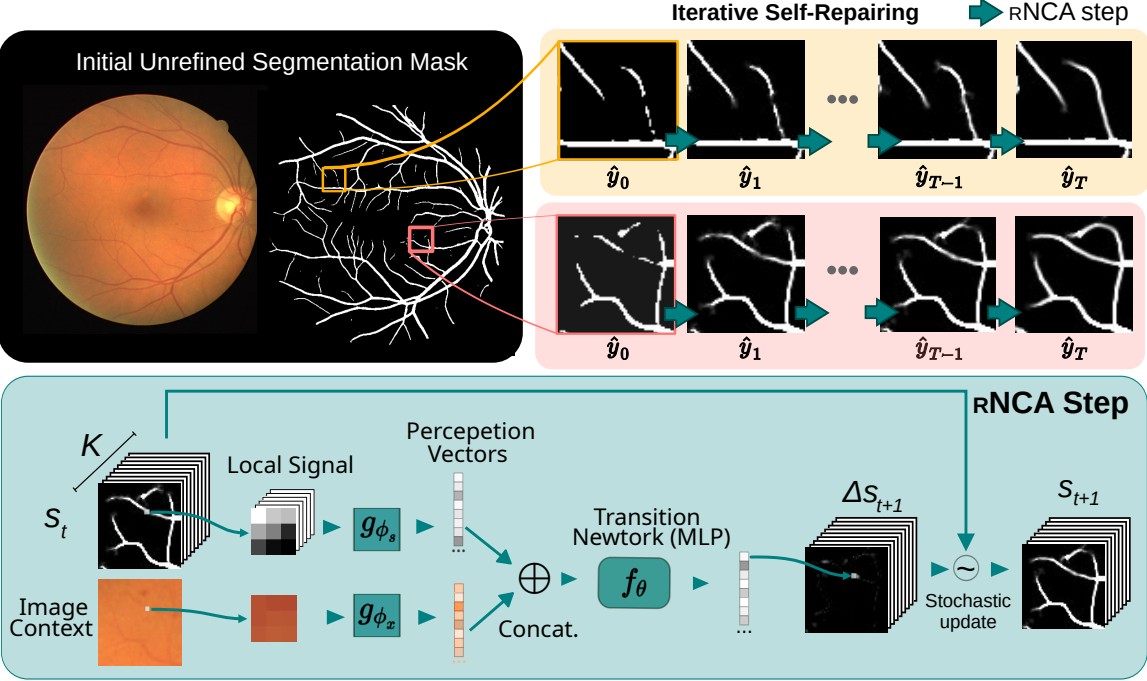

Figure 2: **Overview of RNCA refinement pipeline**. The top panel provides an overview of how RNCA refines an initial mask, which may originate from an external segmentation model or perturbed labels, over $T$ iterative steps. In the bottom panel, we show an overview of a single RNCA step processes the masks state.

keep the refiner tightly close to the original NCA formulation (Mordvintsev et al., 2020) to showcase that the benefits presented here are not due to specializing it to our problem, but it performs as intended without bespoken methods.

## 3.2. Local perception and update rule

At each iteration, every pixel updates its state using only its local neighborhood from the current state $s_t$ and the input image $x$. To extract these local features, we use a learnable perception operator $g_\phi$ composed of two 3×3 convolutional layers for $s_t$ and $x$, formally:

$$g_\phi(s_t, x) = \big[\, g_{\phi_s}(s_t) \,\|\, g_{\phi_x}(x) \,\big], \tag{1}$$

where $g_{\phi_s}(s_t)$ and $g_{\phi_x}(x)$ denote the respective perceive networks and $\|$ denotes the channel-wise concatenation. The output of the perceive network $g_\phi$ are passed to a small learned transition function $f_\theta$, defined as a multilayer perception (MLP), which results in the update:

$$s_{t+1} = s_t + f_\theta(g_\phi(s_t, x)). \tag{2}$$

This formulation enforces strictly local reasoning for state dynamics while allowing information to propagate globally through iterative updates.

Next, masking of *alive pixels* is applied so that only pixels within or near the active regions of $s_t$ and $s_{t+1}$ are allowed to modify the state, leading to growth-pruning dynamics that fix errors while preserving current structures. This contrasts with standard segmentation networks, where each pixel can freely update its predicted value.

### 3.3. Training repair dynamics

The refiner is indirectly trained to repair topological errors by learning dynamics that reconstruct initial perturbed masks, which are either generated by an imperfect segmentation method or by manually corrupting ground-truths (e.g breaking topology and connectivity). This aims to emulate systematic segmentation artifacts.

The training procedure of rNCA is identical regardless of how corrupted masks are generated. Training uses NCA trajectories and employs classic NCA techniques by using a sample pool strategy that continuously samples and updates initial states, forcing the cellular automaton to learn both how to grow the target pattern from a seed and how to maintain it once formed.

More specifically, for the tasks presented in this paper, we maintain a pool of 256 evolving states where at each iteration a minibatch of size $B=32$ is sampled. To expose to the automaton to new perturbed states, we randomly replace $N=2$ entries on the minibatch with new initial states $s_0$ from corrupted masks $y_0$ in the dataset. The rNCA is then unrolled for $T=64$ steps under Eq. (2). Doing so effectively creates the target pattern as an attractor in the system's dynamics. To promote a steady-state solution, the loss is computed at a randomly sampled late step $t^* \sim U(T/2, T)$ using the standard MSE loss between the ground truth $y$ and the prediction at step $t^*$:

$$\mathcal{L} \;=\; |\hat{y}_{t^*} - y\|_2^2. \tag{3}$$

Optimization uses AdamW (Loshchilov and Hutter, 2019) with a learning rate of $10^{-4}$ and decays $\alpha_1=0.9$ and $\alpha_2=0.999$. Notably, no topology-specific losses or additional data augmentations are used during training. For a more detailed explanation on the implementation see Appendix A.

### 3.4. Inference

At test time, the NCA is used as a plug-in refinement operator. Given any predicted mask $\hat{y}_0$ from an arbitrary segmentation model, we initialize latent channels and run a fixed number of update steps to produce a refined mask $\hat{y}_T$. As the model is trained to reach a steady state, the performance is not overly sensitive to the exact number of steps at inference, as long as the number is enough for the repair to occur. To show this, we used the same number of iteration steps for each task. Since the update rule is fully local, performance is independent of mask size and thus can be used in input images of arbitrary resolution.

## 4. Experiments

We evaluate rNCA across three representative classes of topological artifacts of general segmentation methods, as shown in Figure 1.

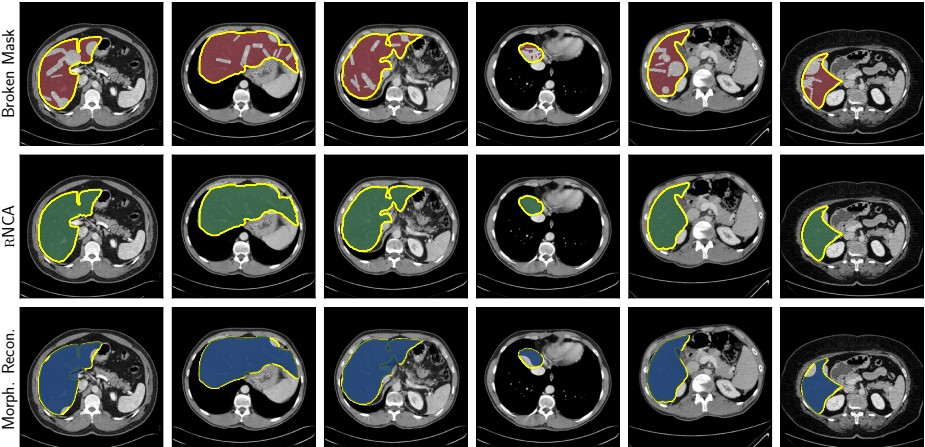

Figure 3: **Liver mask restoration**. Input slice with corrupted mask (top), RNCA output (middle), and morphological reconstruction (bottom), with ground-truth contours in yellow.

First, we reconstruct liver volumes using segmentation with synthetically-generated gaps (4.1). Then, we explore reconnecting thin structures (4.2) exemplified by retinal vessel segmentation masks, while also showing how the method performs on fixing broken rings as can occur in cardiac segmentation in a zero-shot scenario (4.3). We use a shared architecture and hyperparameter configuration across all tasks to demonstrate the robustness of the training procedure across diverse refinement scenarios, with the only differences between them being the data used during training. Nevertheless, we analyze the influence of these parameters and identify failure regimes by providing an ablation study in Appendix C.

To quantify the performance of the methods, we report overlap metrics; Dice and center-line Dice (clDice), together with geometric errors such as average symmetric surface distance (ASSD) and Hausdorff distance (HD), and Betti-number deviations as a proxy for topological correctness (Millson, 1976). Information on the datasets and baseline implementations are available for each experiment in the Appendix B-D.

## 4.1. Fill the Gaps

Holes and missing patches are common artifacts in segmentations of genus-0 structures, where local inconsistencies in pixel-wise classification may lead to anatomically implausible masks. To correct for those, classical morphological operations, such as filling (Soille et al., 1999), are commonly applied and, while effective and simple, become problematic when dealing with objects of non-uniform outline and large degrees of corruptness.

To assess whether RNCA can repair such defects, we construct a synthetic variant of the CHAOS liver dataset (Kavur et al., 2019) by removing regions of varying size and shape from ground-truth masks. The cohort includes 20 CT patients, which we split into training/validation/test sets at the patient level, and from which we create multiple perturbed states to diversion examples of incomplete masks, see Appendix B.

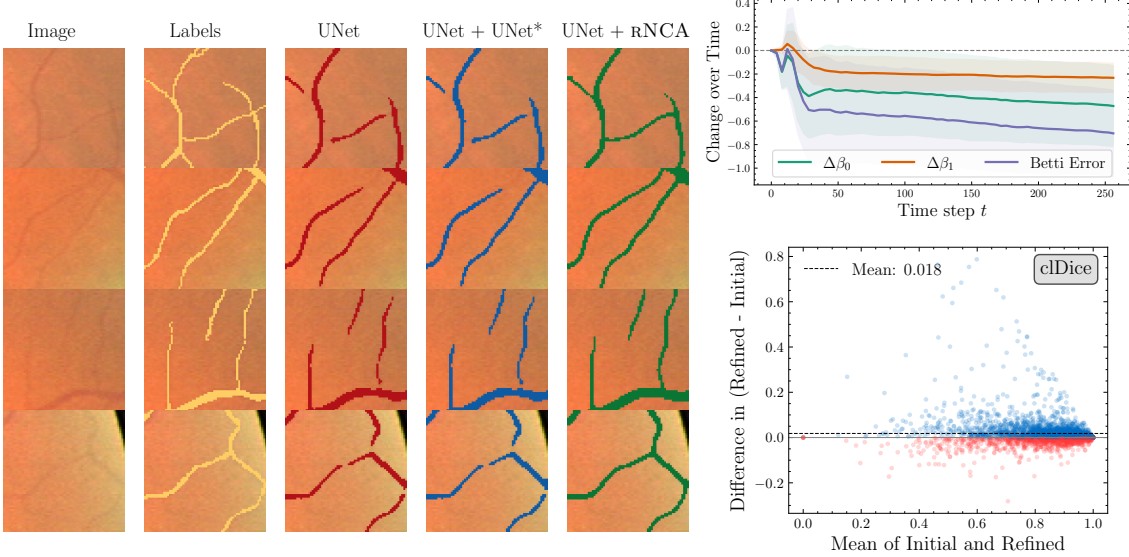

Figure 4: **Retina vessel refinement.** (left) UNet predictions masks with UNet and RNCA refinements. (right, upper) $\beta_0$, $\beta_1$ and Betti error progression during the refinement steps. Here we display the mean and std over all images. (right, lower) Displays a Bland-Altman Plot comparing initial and refined clDice scores of each patch. Plots for Dice and clDice are shown in Appendix C.

As seen in Figure 3, RNCA restores missing regions while preserving the underlying object geometry. Both morphological operations and RNCA improve the Dice score from 0.80 to 0.96 and 0.98, respectively. However, as shown in Figure 3, the morphological operations often over-segment the liver in areas of tight folds or convoluted areas, whereas RNCA learns the anatomical shape, preserving these features. This result is reflected in the distance measures, where RNCA improves ASSD from 0.67 to 0.08 and HD decreases from 10.5 to 4.3, a 58% and 43% improvement compared to morphological operations.

This showcases the benefit of RNCA on situations where context and structural traits can be exploited for improved refinement. Full metrics appear in Appendix B.

### 4.2. Bridge the Gaps

Thin tubular structures are among the most error-prone targets for general segmentation models, as pixel-wise approaches often produce fragment masks at high uncertainty regions. To examine whether RNCA can repair broken connectivity, we explore retinal vessel segmentation on the DRIVE dataset (Staal et al., 2004). Here, unlike in the liver task, broken segments and discontinuities are observed in the UNet segmentation of both healthy and pathological retinal vessels. Therefore, we train RNCA directly on those to expose it to realistic break patterns instead of synthetic perturbations. Evaluation is performed on predictions from both UNet used during training of RNCA and unseen Swin model to assess cross-architecture generalization.

Table 1: Comparison of baseline segmentation models with their refined outputs on DRIVE. Metrics include overlap quality (DICE, clDICE) and topological accuracy ($\beta_0$, $\beta_1$). The best result from each metric is shown in bold.

| Model | DICE ↑ | clDICE ↑ | $\Delta\beta_0$ ↓ | $\Delta\beta_1$ ↓ |
|---|---|---|---|---|
| **Refiners trained and evaluated on UNet** | | | | |
| U-Net | $0.761 \pm 0.027$ | $0.802 \pm 0.028$ | $0.965 \pm 0.291$ | $1.366 \pm 0.535$ |
| + U-Net* | $0.761 \pm 0.027$ | $0.804 \pm 0.027$ | $0.917 \pm 0.291$ | $1.389 \pm 0.535$ |
| + SegFix | $0.759 \pm 0.027$ | $0.800 \pm 0.029$ | $0.957 \pm 0.282$ | $1.372 \pm 0.518$ |
| + SegRefiner | $0.760 \pm 0.026$ | $0.804 \pm 0.029$ | $0.941 \pm 0.279$ | $1.358 \pm 0.524$ |
| + DenseCRF | $0.760 \pm 0.026$ | $0.803 \pm 0.028$ | $0.712 \pm 0.230$ | $1.367 \pm 0.527$ |
| + rNCA | $\mathbf{0.777 \pm 0.029}$ | $\mathbf{0.820 \pm 0.029}$ | $\mathbf{0.532 \pm 0.193}$ | $\mathbf{1.094 \pm 0.485}$ |
| **Refiners trained on UNet, evaluated on unseen Swin model** | | | | |
| Swin | $0.767 \pm 0.026$ | $0.804 \pm 0.025$ | $1.547 \pm 0.393$ | $1.253 \pm 0.529$ |
| + U-Net* | $0.767 \pm 0.026$ | $0.806 \pm 0.025$ | $1.408 \pm 0.393$ | $1.291 \pm 0.529$ |
| + SegFix | $0.765 \pm 0.026$ | $0.802 \pm 0.028$ | $1.419 \pm 0.352$ | $1.265 \pm 0.519$ |
| + SegRefiner | $0.766 \pm 0.025$ | $0.806 \pm 0.026$ | $1.485 \pm 0.391$ | $1.255 \pm 0.517$ |
| + DenseCRF | $0.767 \pm 0.026$ | $0.806 \pm 0.023$ | $0.902 \pm 0.213$ | $1.259 \pm 0.523$ |
| + rNCA | $\mathbf{0.776 \pm 0.032}$ | $\mathbf{0.816 \pm 0.030}$ | $\mathbf{0.646 \pm 0.187}$ | $\mathbf{1.122 \pm 0.475}$ |

Figure 4 qualitatively demonstrates that rNCA effectively bridges gaps in fragmented vein segmentations, and that the biggest improvement occurs in the initial steps and maintains a steady state from there. It also compares initial and refined clDice scores of individual patches, and suggests that, on average rNCA provides modest improvements while maintaining stability across all initial scores. Consistent with these observations, Table 1 shows that rNCA increases both Dice and clDice, and notably reduces Betti-number deviations from the base-method predictions. Using the U-Net as a refiner results in minor changes from the initial predictions. This is likely due to the model getting stuck in a local minima early due to its reliance on the feeded initial predictions. Despite being trained solely on UNet outputs, the refiner generalizes well and similarly improves masks generated by Swin. To benchmark against prior refinement methods, we compare rNCA with SegFix (Yuan et al., 2020b), SegRefiner (Wang et al., 2023), DenseCRF (Krähenbühl and Koltun, 2011), and a conditional UNet trained to refine the initial guess, all of which are trained on the same dataset with UNet predictions until convergence, see Appendix C.

rNCA outperforms all baselines across metrics and base models. In particular, rNCA has large improvements in topology performance, showing its ability to indirectly learn the topological features from the training dataset. We note that, unlike rNCA's iterative refinement, these approaches rely on a single-pass correction, which may limit their ability to handle challenging predictions.

### 4.3. Close the Gap

Breaks in the myocardial rings are a frequent failure mode in cardiac MRI segmentation and directly affect perimeter-based clinical measurements (Wyburd et al., 2024). To study the applicability of RNCA on such scenarios where contour completeness is broken, we use *ACDC* myocardium masks (Bernard et al., 2018) and generate perturbations via thinning, opening, and thickening on the ground truth. Unlike the retinal vessel tasks, most of the predictions from segmentation models are already topologically correct, so in order to avoid overcorrection from RNCA, we include unperturbed ground truths into the dataset, see Appendix D for more details.

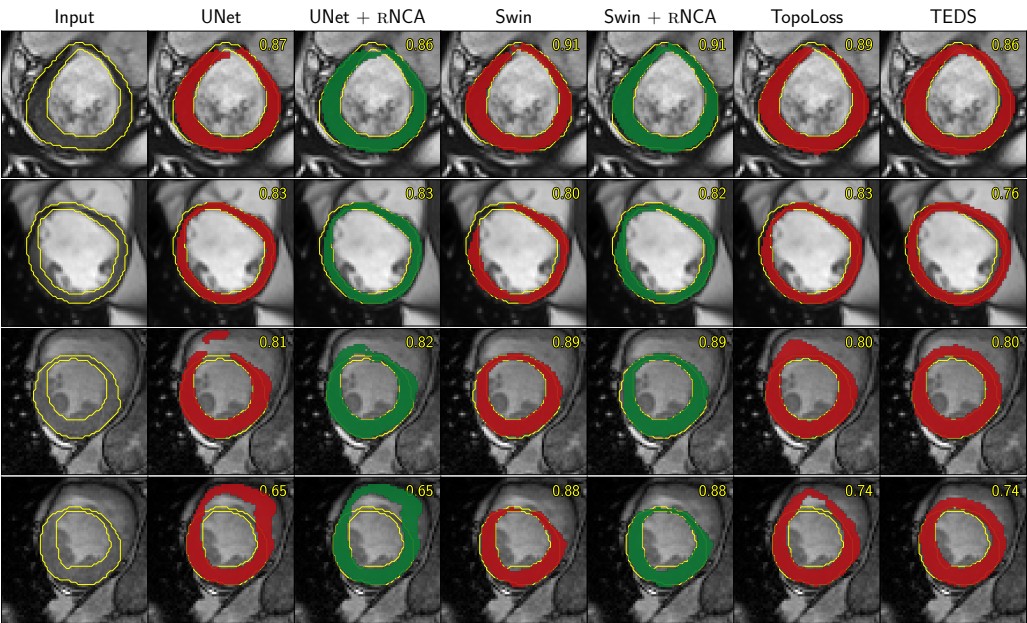

Figure 5: **Myocardium ring correction.** Predictions before and after RNCA, alongside TopoLoss and TEDS. Dice scores shown in yellow; bottom row illustrates a failure case (UNet+RNCA), with more shown in Appendix D.

We evaluate RNCA on predictions from independently trained UNet and Swin models, each exhibiting distinct topological errors, with implementation described in Appendix D. Results are then reported both on the full test set in Table 2 as well as on those cases where the ring topology was not preserved in the initial segmentation (Table 3).

Refining both models' predictions with RNCA improves segmentation performance across all metrics, with particularly strong gains on cases that initially had non-closed loops (Table 3). We further compare RNCA's with two topology-aware approaches: a topological loss function TopoLoss (Hu et al., 2019) and a topology-preserving deformation model, TEDS-Net (Wyburd et al., 2021), explicitly designed for myocardium heart segmentation.

Figure 5 shows qualitative examples of myocardium repair. Here, RNCA is shown to repair gaps across the networks. Although TEDS-Net has 100% topology accuracy, it often

produces overly segmented, misaligned rings (see Appendix D), whereas, RNCA is able to join disconnected regions following the real contours. This figure also highlights a limitation of the method: as RNCA is a local refinement process, if an initial mask is particularly poor (bottom row and Appendix D), it can not effectively repair it.

Table 2: Performance on all myocardium test slices (100 slices). RNCA preserves global segmentation accuracy and Betti-number stability.

| Model | DICE ↑ | ASSD ↓ | $\Delta\beta_0 \downarrow$ | $\Delta\beta_1 \downarrow$ |
|---|---|---|---|---|
| U-Net | $0.868 \pm 0.155$ | $0.81 \pm 0.90$ | 0.04 | 0.07 |
| + RNCA | $0.871 \pm 0.145$ | $0.83 \pm 1.15$ | 0.03 | 0.05 |
| Swin | $0.855 \pm 0.173$ | $0.79 \pm 0.72$ | 0.09 | 0.12 |
| + RNCA | $0.859 \pm 0.166$ | $\mathbf{0.75 \pm 0.53}$ | 0.02 | 0.06 |
| TopoLoss | $\mathbf{0.876 \pm 0.150}$ | $0.83 \pm 1.26$ | 0.02 | 0.06 |
| TEDS | $0.848 \pm 0.140$ | $0.94 \pm 0.75$ | **0.00** | **0.00** |

Table 3: Performance on slices where the predicted myocardium ring is broken. The upper block corresponds to U-Net failures and the lower block to Swin failures.

| Model | DICE ↑ | HD ↓ | ASSD ↓ | Topo. ↑ |
|---|---|---|---|---|
| **UNet broken subset (8 slices)** | | | | |
| UNet | $0.490 \pm 0.345$ | $13.48 \pm 6.30$ | $2.96 \pm 2.30$ | 0/8 |
| + RNCA | $\mathbf{0.517 \pm 0.318}$ | $12.11 \pm 8.60$ | $3.27 \pm 3.34$ | 2/8 |
| TopoLoss | $0.515 \pm 0.336$ | $10.47 \pm 6.16$ | $3.53 \pm 3.34$ | 3/8 |
| TEDS | $0.499 \pm 0.293$ | $\mathbf{8.69 \pm 4.96}$ | $\mathbf{2.57 \pm 1.86}$ | **8/8** |
| **Swin broken subset (13 slices)** | | | | |
| Swin | $0.539 \pm 0.329$ | $7.09 \pm 4.22$ | $2.01 \pm 1.65$ | 0/13 |
| + RNCA | $0.569 \pm 0.325$ | $\mathbf{5.98 \pm 4.07}$ | $\mathbf{1.63 \pm 1.16}$ | 8/13 |
| TopoLoss | $\mathbf{0.611 \pm 0.293}$ | $8.10 \pm 5.79$ | $2.65 \pm 2.87$ | 7/13 |
| TEDS | $0.600 \pm 0.264$ | $6.91 \pm 4.57$ | $2.08 \pm 1.61$ | **13/13** |

## 5. Discussion

Across three structurally distinct repair tasks, RNCA has consistently improved topological correctness while minimally affecting already good predictions. We have seen that training both on corrupted labels and on model predictions yields effective repair dynamics that generalize across architectures. We have also shown that, due to its simplicity, the method is easy to port to different datasets without having to fine tune its hyperparameters. These

results showcase that local refinement is sufficient to correct a broad range of structural failures present in popular segmentation models.

The experiments also show why NCA naturally fits the refinement task. Corrections are applied only where the state dynamics detect inconsistencies, much similar to biological systems repair mechanisms, while limiting the risk of over-refinement. This trait is particularly relevant in the setting of medical applications, where excessive and anisotropic mask postprocessing can produce large mask alterations that are often undesirable.

Nevertheless, the focus on local dynamics also presents its limitations. For too broken initial states that are unseen during training, rNCA is not able to recover the ground truth as they require long-range reasoning. When the image is too ambiguous, local context may be insufficient to learn the correct dynamics. An example of this is shown in Supplementary Figure 7, where the initial predictions and corresponding image do not contain enough information for the rNCA to effectively improve the segmentations. Furthermore, despite the small network size, the NCAs training is more complex due to its iterative nature. It has to not only learn from the initial masks, but also learn how to correct and balance its own predictions into a steady state solution, which results in slower training times than single pass methods.

A practical advantage of rNCA is that it offers a drop-in mechanism that does not need to interact with the base model training process. Its smaller parameter count ($\sim 12k$, see Appendix Table 5) and architecture makes it portable and ideal to be deployed in edge devices. (Kalkhof et al., 2023), where inference computational requirements, both in runtime and memory consumption, is paramount. This rNCA an attractive refinement layer for medical segmentation pipelines. Its steady-state solution preserves accurate segmentation while correcting those that are suboptimal.

## 6. Conclusion & Future Research

This work introduces refinement Neural Cellular Automata as a simple, data-driven mechanism for repairing segmentation masks. By learning local repair dynamics from the data, rNCA improves structural correctness across tasks with minimal integration effort. The approach provides an attractive alternative to topology-aware training strategies and simplifies the refinement approach by using already explored techniques in the field of growth dynamics. Looking forward, the main challenges lie in extending the method beyond proof-of-concept scenarios and into more integrated pipelines.

In this work, we have shown the rNCA on three binary 2D segmentation tasks; however, extending the approach to 3D and multi-class segmentation remains an exciting direction for future research. Addressing these would result in an off-the-shelf refinement method for a large domain of tasks.

Overall, this study shows that NCA offer a natural and surprisingly effective fit for segmentation refinement. They are simple, easy to deploy, and capable of repairing errors that compromise downstream reliability without task-specific customization of the refiner, opening exciting new directions in refinement solutions for clinical settings.

## Acknowledgments

MS, JP and MB are supported by the European Union's Horizon Europe under Grant No. 101080983. MKW is supported by Danish Data Science Academy, which is funded by the Novo Nordisk Foundation (NNF21SA0069429). BI is supported by the Novo Nordisk Foundation under Grant NFF20OC0062056.

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

## Appendix A. Model Architecture

The refiner follows a Neural Cellular Automaton formulation in which each pixel holds a $K$-dimensional state vector. In rNCA the first channel encodes the visible mask, while the remaining $K - 1$ channels act as latent variables. We use $K = 16$ unless otherwise stated. During inference, the visible state is initialized from the predicted mask, and all latent channels start at zero.

Local information comes from two sources. The current state $s_t \in \mathbb{R}^{H \times W \times K}$, where $H$ & $W$ denotes the image size, is processed by a learnable perception operator $g_{\phi_s}(s_t)$ : $\mathbb{R}^{H \times W \times K} \to \mathbb{R}^{H \times W \times K}$. It applies a small set of learnable spatial filters, initialized as the original identity, Sobel (x and y). We chose to have all filters learnable to give the network complete freedom. In parallel, the image $x \in \mathbb{R}^{H \times W \times C_x}$ with $C_x$ denoting the number of image channels, is processed by a separate learnable convolutional layer $g_{\phi_x}(x)$ : $\mathbb{R}^{H \times W \times C_x} \to \mathbb{R}^{H \times W \times N}$ that extracts local appearance cues at the same spatial resolution as the mask. We limit this layer to be a $3x3$ convolutional layer whose output matches the dimension of $g_{\phi_s}(s_t)$ output ($N$=64 for $K$=16). This way the update step receives equal contribution from the image and the state. The outputs of these two components are concatenated and passed to a lightweight transition network, implemented as a two-layer MLP with ReLU activations and a hidden width of 128. The update rule is additive, akin to an Euler integration step with $\Delta t = 1$:

$$s_{t+1} = s_t + f_\theta(g_\phi(s_t, x)). \tag{4}$$

A pixel is considered *alive* if its visible value or that of any of its 8-neighbors exceeds 0.1, before and after the update step. Formally, let $\Omega \subset \mathbb{Z}^2$ be the set of pixel coordinates. The 8-neighbourhood of a pixel $v = (i, j) \in \Omega$, including $v$ itself, is defined as:

$$N_8^+(v) = \{(m, n) \in \Omega \mid |m - i| \leq 1, \ |n - j| \leq 1\}.$$

Let $\alpha = 0.1$. The masking rule for $s_{t+1}(v)$, i.e. K-state vector of a pixel $v$ is:

$$s_{t+1}(v) := \begin{cases} s_{t+1}(v), & \text{if } \max_{q \in N_8^+(v)} \hat{y}_{t+1}(q) \geq \alpha \text{ and } \max_{q \in N_8^+(v)} \hat{y}_t(q) \geq \alpha, \\ 0, & \text{otherwise.} \end{cases}$$

Only alive pixels may update the visible channel, while all pixels update latent channels at every step. To improve robustness and following the original work, each pixel independently skips its update with probability $\tau = 0.5$, introducing controlled stochasticity into the dynamics.

### A.1. Training Process

The refiner learns to reconstruct clean labels from inputs that contain realistic segmentation errors. The source of these errors differs across tasks. For datasets such as ACDC and the synthetic liver experiment, we generate corrupted masks directly from the ground-truth labels. The reason behind this being either to have a more controlled topological repair study or because topological errors, despite being critical, are less prominent on the resulting predictions, and thus the refiner does not focus on those artifacts. These corruptions are

created using simple morphological perturbations, such as thinning, small deletions, or mild opening and closing operations, which emulate typical segmentation failures.

For the retinal vessel segmentation task, the situation is different. Thin structures often break in practice, and U-Net predictions reflect these characteristic failure modes. To expose the refiner to such realistic errors, we train it directly on U-Net predictions paired with the ground-truth labels. Although the model is trained on U-Net outputs, we also evaluate it on top of Swin predictions to show that the learned dynamics generalize across architectures. The training procedure is identical regardless of how corrupted masks are generated.

## Appendix B. Experiments: Liver (CHAOS-CT)

**Dataset**   Each CT slice is cropped to $128 \times 128$ around the liver. Corruptions remove or modify regions to create holes, missing patches, or eroded boundaries. The dataset is split by patient into 14 training, 3 validation, and 3 test subjects. Each patient contains $\sim 10$ liver slices, which we augment and generate $3\times$ perturbations of initial states, each with randomly removed chunks as exemplified in Figure 3. There removed sections are rectangles or arbitrary sizes as well as circles, which we allow to overlap to create more variate of holes.

**Quantitative Results on Liver Filling Task**   For completeness, reports the full set of liver-filling results, including mean and standard deviation for all metrics. These values complement the main text by showing the variance across test slices and the relative performance of the initial masks, morphological reconstruction, and rNCA.

Table 4: Liver filling task: full metrics with mean $\pm$ std.

| Method | HD $\downarrow$ | ASSD $\downarrow$ | DICE $\uparrow$ |
|---|---|---|---|
| Initial mask | $10.51 \pm 3.68$ | $0.673 \pm 0.887$ | $0.798 \pm 0.145$ |
| Morph. reconstr. | $7.57 \pm 4.28$ | $0.190 \pm 0.790$ | $0.958 \pm 0.090$ |
| **rNCA** | $3.66 \pm 3.20$ | $0.094 \pm 0.779$ | $0.983 \pm 0.089$ |

## Appendix C. Experiments: Retinal Vessels (DRIVE)

**Dataset and Training**   The DRIVE (Staal et al., 2004) dataset consists of 20 images, developed to provide a standardized benchmark for evaluating retinal blood vessel segmentation methods in the context of diabetic retinopathy screening. Due to its small dataset size we randomly split the data 0.8:0.2 across three folds and report the average validation metrics, as done in prior work see (Hu et al., 2019). From the full size image we extract patches of size 64x64 for the NCA training using a sliding window with 0.25 overlap. We disregard empty initial predictions as there is no signal to refine. We do this for each fold and each baseline model we try to refine. This creates $\tilde{4}000$ patches for all 20 images.

**Baselines**   The initial predictions are generated using a UNet and Swin. The UNet is configured with five encoder levels with channel sizes $(16, 32, 64, 128, 256)$ and downsampling strides of 2 at each level. Each block contains 2 residual units with a $5\times5$ kernel, and the decoder uses $3\times3$ upsampling kernels. The Swin-UNETR we configure feature size 48, where

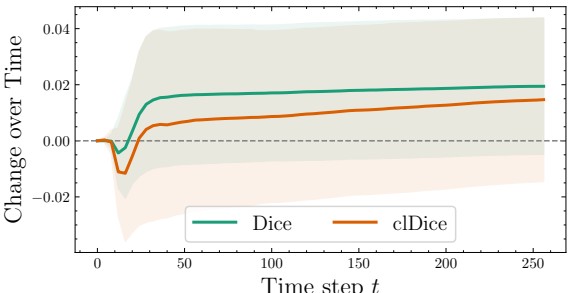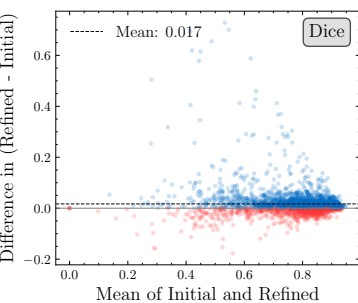

Figure 6: **Retina vessel refinement.** (left) clDice and Dice progression during the refinement steps. Here we display the mean and std over all images. (right) Displays a Bland-Altman Plot comparing initial and refined Dice scores of each patch.

the encoder uses Swin Transformer v2 blocks with patch size 2, depths 2 at each level, number of heads of 3, 6, 12, 24 respectively and a window size of 7. Trained with a Dice loss and AdamW with a learning rate of $10^{-2}$

Each model is trained to predict patches of size 128x128, randomly sampled during training. We also apply random flips, rotation, zoom, and random grayscale blending to the samples before cropping. During inference, we predict the full size image using a sliding window with 0.5 overlap and Gaussian averaging of the predicted logits across the patches. The model with the highest Dice score on the full sized validation images is selected for each fold and used to create the initial predictions that we will refine.

Furthermore, we use UNet and test its ability to refine the initial prediction and turn the prediction problem into a residual learning task. Now, trained on the same 64x64 patches as rNCA, the UNet receives the concatenation of initial prediction and image. This is to see how well the UNet compares in the same setting.

We compare rNCA to three popular segmentation refinement methods: SegFix (Yuan et al., 2020b),SegRefiner (Wang et al., 2023) and DenserCRF (Krähenbühl and Koltun, 2011). Each refiner was trained on the same dataset as rNCA, until convergence, using their publicly available implementations for Segfix[1], SegRefiner[2] and DenserCRF [3].

**Additional Results**  Complementary to the results shown in Section 4.2 we add some additional results for completeness. Figure 6 (left) shows the same convergence behaviour for clDice and Dice during inference. Note there is a small drop in performance in the early states, which we attribute to an initial exploration phase, as no loss is computed for $T < 32$. Furthermore, Figure 6 (right) shows that the Dice scores follow the same pattern observed for clDice, supporting the consistency of rNCA 's behaviour at the patch level.

**Ablation Experiments**  To better understand which components drive the refinement behavior, we perform an ablation study on this task. We systematically vary the number

---

1. https://github.com/openseg-group/openseg.pytorch

2. https://github.com/MengyuWang826/SegRefiner

3. https://github.com/heiwang1997/DenseCRF

of state channels $K$, the pool-replacement rate $N$, and the number of unrolled steps $T$, and additionally compare variants with and without an image encoder, as well as supervision from perturbed labels versus model predictions. All variants are trained with the same protocol and evaluated on DICE, clDICE, and Betti number deviations. As summarized in Table 6, this setup isolates the effect of architectural capacity, state refresh frequency, rollout depth, and supervision source on the ability of RNCA to correct fragmented vessels. Here we trained the model to repair vessels perturbed by morphological operations. Each label randomly gets 75% of their thin structures removed by erosion, and returns the mask with most thin structures removed. This allows for a more controlled environment to analyse the models behavior. Ablation is done on a random fold, different to the training folds. We can see by decreasing the rollout depth too much (e.g. $T = 16$) or replace the whole batch $N = B$, the model does not achieve a steady state. Both configuration limit (or prevent completely) the progression of samples and hence lead to instability during inference. Furthermore, limiting model capacity $K = 1$ decreases performance, which tells us that the hidden channels learn important information about the growth directions. Interestingly, increasing the hidden channels further did not contribute any performance gains, presumably due to redundant information saturating the system. Increasing roll-out time ($T = 128$) furthermore did not seem to improve the performance much as RNCA already reaches a steady state solution with less memory cost. Lastly, we tried a formulation of RNCA (Static) close to the original NCA, in which both image channels and states are passed through pre-defined non-learnable filters and concatenated afterwards. This limited performance significantly, most likely due to limited information from the image influencing the update step.

**Runtime analysis** A key requirement for a segmentation model is that it is able to perform well on edge devices. Here we should that RNCA runtime on refining 64x64 patches compared to the other refiners, as well as its parameter count, Table 5. Because refinement is iterative, RNCA uses 64 update steps per image (the repair phase shown in Figure 4) Despite this, its end-to-end runtime is on par with or faster than most learned refiners, while its parameter count remains extremely small (12,048), comparable to SegRefiner and orders of magnitude lighter than UNet- or SegFix-based refinement. This makes RNCA one of the most computationally efficient refinement mechanisms among the evaluated methods.

## Appendix D. Experiments: Myocardium

**Dataset and Training** For the myocardium segmentation, we used the ACDC dataset from (Bernard et al., 2018). This dataset contains two labelled 3D cardiac scans from 100 patients. From each scan, we extracted 2 myocardium-containing slices, cropped to 144 by 208 pixels. The dataset was split by patient into 85 (850 slices) training, 15 (150 slices) validation, and 10 (100 slices) for testing.

To expose the refiner to possible topological artifacts, we generated initial states by performing multiple morphological augmentations on the ground truth. Erosion and dilation were used to thin or thicken boundaries, sometimes breaking the topology. This is critical for ACDC dataset where different thickness may be expected at different arc segments and we require the refiner to be able to handle both types of operations. Additional, binary opening was applied to promote the cases of ring breaking. Lastly, as the initial predictions can be

Table 5: Runtime and parameter comparison for CPU and GPU execution.

| CPU Results | | | |
|---|---|---|---|
| **Method** | **Time (s)** | **Time/Image (ms)** | **Parameters** |
| DenseCRF | $0.0195 \pm 0.0001$ | 3.90 | N/A |
| SegFix | $0.0246 \pm 0.0013$ | 4.91 | 1,931,392 |
| SegRefiner | $0.0023 \pm 0.0000$ | 0.46 | 9,601 |
| UNet | $0.0213 \pm 0.0000$ | 4.27 | 3,976,476 |
| RNCA | $0.376 \pm 0.177$ | 75.20 ($1.17\,\mathrm{ms/iter}$) | 12,048 |
| **GPU Results** | | | |
| **Method** | **Time (s)** | **Time/Image (ms)** | **Parameters** |
| SegFix | $0.0133 \pm 0.0164$ | 2.66 | 1,931,392 |
| SegRefiner | $0.0011 \pm 0.0010$ | 0.23 | 9,601 |
| UNet | $0.0102 \pm 0.0001$ | 2.05 | 3,976,476 |
| RNCA | $0.0124 \pm 0.0001$ | 2.48 ($0.039\,\mathrm{ms/iter}$) | 12,048 |

Table 6: Results from the ablation study assessing how architectural and training choices affect the refiner. The table compares different state channel sizes $K$, pool–replacement rates $N$, and loss timestep $T$, along with the impact of adding/removing a learnable image encoder and training on perturbed labels instead of predictions. RNCA is initialized with $(K = 16, N = 2, T = 64)$ The comparison highlights which configurations yield more reliable repair of thin structures.

| | **DICE** $\uparrow$ | **clDICE** $\uparrow$ | $\Delta\beta_0 \downarrow$ | $\Delta\beta_1 \downarrow$ |
|---|---|---|---|---|
| (Perturbed Labels) | 0.8518 | 0.7919 | 1.1801 | 2.0233 |
| RNCA | 0.9088 | 0.8799 | 0.2306 | 0.9197 |
| RNCA ($K = 1$) | 0.8498 | 0.7914 | 1.0984 | 2.0622 |
| RNCA ($K = 32$) | 0.8930 | 0.8513 | 0.5052 | 1.4378 |
| RNCA ($N = B/2$) | 0.8971 | 0.8573 | 1.0531 | 1.3951 |
| RNCA ($N = B$) | 0.2073 | 0.2353 | 0.3653 | 109.044 |
| RNCA ($T = 16$) | 0.6113 | 0.5497 | 47.3303 | 3.6218 |
| RNCA ($T = 128$) | 0.9109 | 0.8812 | 0.4573 | 1.1049 |
| RNCA (Static) | 0.8879 | 0.8327 | 3.8394 | 1.3135 |

good and we do not want to over refine good initial states, we included unperturbed ground truth masks. These targeted distortions created a consistent spectrum of defects, gaps, oversegmented regions, undersegmented regions, and isolated artifacts that the refinement model was explicitly trained to repair.

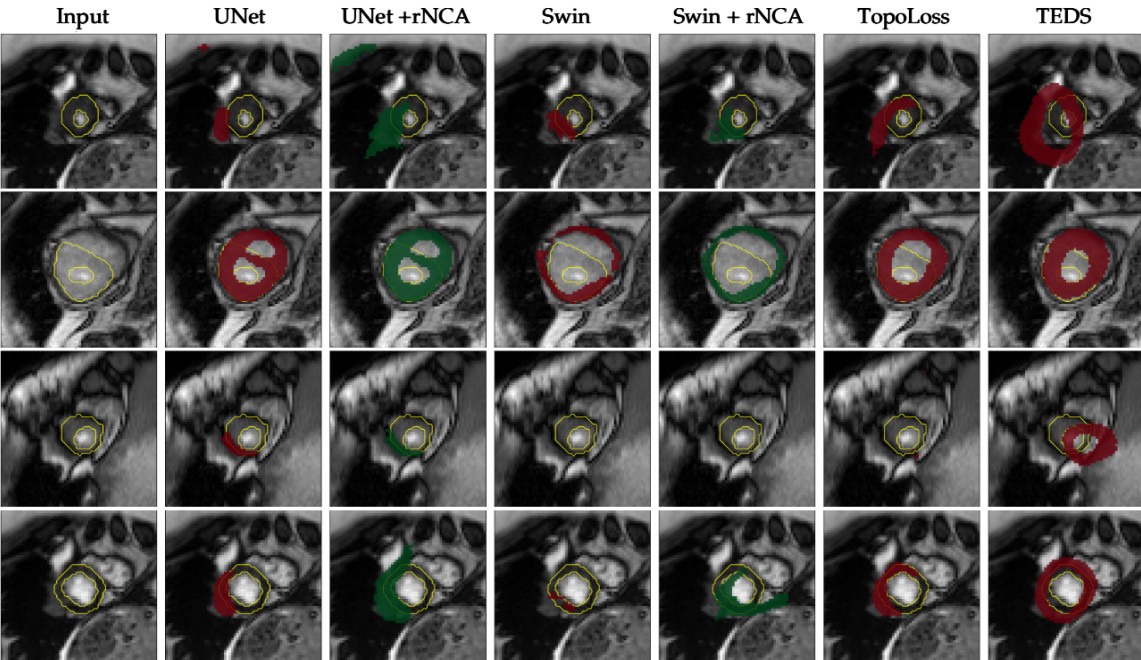

Figure 7: Examples of initially poor masks which RNCA is unable to correct.

**Baselines** For the base segmentation models, we used the published predictions from transformer based Swin-UNETR (Cao et al., 2022) and U-Net (Ronneberger et al., 2015). For topology-based comparisons, we likewise used the published predictions from two additional methods. The first is TopoLoss (Hu et al., 2019), a topological loss function that aligns the persistent homology of predicted probability maps with that of the ground-truth labels. The second is TEDS-Net (Wyburd et al., 2021), a spatial transformer network that learns a topology-preserving deformation field and warps a prior shape of known topology to each input. TEDS-Net requires explicit prior knowledge of the anatomical topology. All model implementations and predictions follow the descriptions provided in (Wyburd et al., 2024).

**Additional Results** Figure 7 shows examples of poor initial predictions from the base segmentation models that RNCA is unable to correct. Further, it shows that despite TEDS-Net achieving 100% correct topology, often the segmentations are misaligned.

