# OpenReview forum: "rNCA: Self-Repairing Segmentation Masks"
_MIDL.io/2026/Conference — MIDL 2026 Poster_

### Official Review · Reviewer_LMFs · 2025-12-26

**Confidence:** 4
**Preliminary Rating:** 3

**Summary:**

The paper proposes rNCA, a module based on neural cellular automata that learns local, iterative update rules to “self-repair” fragmented or topologically incorrect segmentation masks using image context. Learned on corrupted masks paired with ground truth, the system treats correct anatomy as a stable attractor and progressively reconnects broken regions, removes spurious fragments, and closes gaps without retraining the base segmentation model.

**Strengths:**

1. rNCA may easy to integrate into existing clinical pipelines with minimal engineering effort.
2. It should indicate that rNCA learns task-level repair rules rather than model-specific artifacts.
3. Across experiments, rNCA reduces Betti-number errors.
4. rNCA contains about 12k parameters, fewer than typical UNet-based refiners.

**Weaknesses:**

1. rNCA relies primarily on local dynamics and therefore struggles to recover cases with large missing regions or when the initial prediction deviates substantially from the target anatomy. As a result, the method exhibits a strong dependence on the quality of the baseline segmentation.
2. The convergence behavior is highly sensitive to hyperparameters such as rollout length, pool replacement rate, and the number of state channels, which may hinder reproducibility and make the method difficult to adopt in practice.
3. It remains unclear how well the approach generalizes to more diverse segmentation scenarios, including multi-class settings, natural images, or highly cluttered structures.
4. The number of refinement iterations must be manually selected and may vary across tasks and datasets. This introduces additional latency and hyperparameter tuning compared with one-shot refinement methods.

**Detailed Comments:**

Please see above

**Justification Of The Preliminary Rating:**

While the proposed idea shows some promise, the current experimental evidence and discussion are not sufficiently strong or comprehensive to fully support acceptance, resulting in a borderline recommendation.

**Questions To Address In The Rebuttal:**

Please see the section on weaknesses

---

> ### Author Response · Authors · 2026-01-23
>
> We thank the Reviewer for their time in this response.
>
> # Dependence on initial predictions
> We agree with the Reviewer that rNCA is strongly dependent on the quality of the initial segmentation. This is an inherent limitation of refinement-based approaches that rely on local update rules.
> When the initial prediction is particularly poor, e.g. missing large regions, and such extreme examples are not seen during training, the rNCA is insufficient to recover the correct structure. We show examples of this in Supplementary Figure 7, and now address it explicitly in the discussion:
>
> “Nevertheless, the focus on local dynamics also presents its limitations. For too broken initial states that are unseen during training, rNCA is not able to recover the ground truth as they require long-range reasoning. When the image is too ambiguous, local context may be insufficient to learn the correct dynamics. An example of this is shown in Supplementary Figure 7, where the initial predictions and corresponding image do not contain enough information for the rNCA to effectively improve the segmentations.”
>
> Additionally, while this dependence on the baseline segmentation is a limitation, in practice, many out-of-the-box segmentation methods already produce strong initial predictions that can be further refined with rNCA. This enables a general refinement strategy and reduces the need for specialised, task-specific models. Further, this effect can be reduced by training on examples with similar types of errors, as shown in the vessel experiment (section 4.2). As the rNCA was trained using initial states from the U-Net, the train-time errors are consistent with the errors seen at test time, as shown by the Bland–Altman plots in Figure 4, where Dice performance improves in most cases without dramatic degradation.
>
>
> # Network Convergence
> Although the training dynamics of the NCA are non-trivial and relatively underexplored, our empirical results indicate that rNCA can be trained reliably using a fixed set of hyperparameters. Across all datasets and experiments, we intentionally used the same hyperparameter configuration to demonstrate the simplicity and robustness of the approach, which consistently produced stable convergence and competitive results.
>
> However, we do recognise the importance of these parameters, so to explore in what settings the methods break down, we performed an ablation study (Supplementary Table 6). To make this clear to the reader, we now refer to this experiment directly in the main text, Section 3:
>
> “We use a shared architecture and hyperparameter configuration across all tasks to demonstrate the robustness of the training procedure across diverse refinement scenarios, with the only differences between them being the data used to train. To analyze the influence of these parameters and identify failure regimes, we additionally provide an ablation study in Appendix C.”
>
> # rNCA generability to new segmentation cases
> We agree that there is a diverse range of segmentation scenarios the rNCA has yet to be tested on. While application to more datasets/scenarios is beyond the scope of this paper, we believe the proposed formulation is not inherently restricted to binary 2D segmentation. In particular, multi-class segmentation could be addressed by allocating separate channels, enabling class-wise refinement with only a modest increase in memory cost. This is in fact how the original NCA does it to regrow RGB patterns (i.e. with three visible channels). This design may be especially relevant for cluttered or overlapping structures, where independent refinement dynamics could be beneficial. However, this was not the focus of this paper, as we first wanted to introduce the method and show the types of topology errors it could fix.
>
> As these are all relevant open and exciting questions for future work, we have expanded our future work section to include them.
>
>
> # Choice of number of iterations at inference
>
> As correctly pointed out, rNCA is an iterative method and thus the number of refinement iterations must be specified at inference time as a hyperparameter. However, rNCA is trained to converge to a steady-state solution rather than to produce a fixed-time output, resulting in a relaxed dependence on the exact number of iterations once the rollout is sufficiently long. In all experiments, we use the same number of refinement steps and observe stable behaviour across tasks, indicating limited sensitivity to this choice.
>
> Additional iterations beyond convergence do not significantly affect the output. An explicit convergence-based stopping criterion could further remove the need for manual selection and is a natural extension of the proposed framework.
>
> We have added a clarification in Section 3.4 to make this behaviour explicit

---

> ### Comment · Area_Chair_kBWM · 2026-02-02
>
> Reviewer LMFs,
>
> Could you please review the rebuttals and share your feedback? thanks

---

### Official Review · Reviewer_3hAG · 2026-01-09

**Confidence:** 5
**Preliminary Rating:** 5

**Summary:**

This paper introduces a refinement mechanism that employs local, iterative updates guided by image context to improve segmentation masks. By utilizing local information, it enhances structural accuracy across three segmentation tasks: filling, re-connection, and closing.
The experiments conducted in the paper demonstrate that the method naturally fits the refinement task.

**Strengths:**

- Using a method that can reconstruct the missing parts of images by filling, re-connection, and closing.
- Learning from neighbouring pixels and applying Nonlinear Cell Automata is particularly exciting, as NCA itself is inspired by biological processes such as tissue development and the organization of living organisms.

**Weaknesses:**

- The pipeline is composed of multiple stages, each designed to perform a specific function. It may add to the overall complexity of the system, making it more difficult to optimize effectively, especially for the challenging cases.
- As it uses learning from neighbouring pixels, and medical images are usually contain noises, the error propagation can occur across the various stages of the pipeline.

**Detailed Comments:**

- Incorporating certain sections from the appendix into the methodology section of the paper.
- Providing further clarification of the methodology by adding additional figures that illustrate the system's structure.
- Incorporating challenging examples where the method cannot fully reconstruct the object's outline.

**Justification Of The Preliminary Rating:**

Biologically inspired methods have always been a fascinating approach to tackling complex problems. The proposed method utilizes one such approach to address a challenging issue in segmentation. While the method presents three distinct pathways, there appears to be potential for improvement, where all of these pathways could be integrated into a single, unified method.

**Questions To Address In The Rebuttal:**

- Representing a more detailed explanation of the methodology by including extra figures that demonstrate the system's structure.
- Introducing difficult examples where the method fails to completely reconstruct the outline.

---

> ### Author Response · Authors · 2026-01-23
>
> We thank the Reviewer for their positive review. We have followed their guidance and updated the methods for clarity and added further discussion on the limitations.
>
> # Updated methods section
>
> We have updated Figure 2, to show (top panel) an overview of the iterative refinement process and, below, a more detailed overview of the rNCA structure. We have also improved the figure caption and added to the methods section to improve clarity. We have also expanded the training description to describe the NCA training process.
> # rNCA failure modes
>
> We agree with the Reviewer about the importance of including failure modes of the model. Given a reasonable initial mask, it performs well. This is shown in the Bland Altman plots (Figure 4), where in the vast majority of cases the Dice performance is improved from the initial predictions (shown in blue), with no dramatic decrease in performance. Further, even in cases with poor initial predictions, the model is able to improve upon them, as it has seen examples during training. However, if the initial predictions are poor and unseen during training, the refinement method can struggle to improve it. We showed this in Supplementary Figure 7, however, to make this explicitly clear in the text, we have added the following to our discussion:
>
> “Nevertheless, the focus on local dynamics also presents its limitations. For too broken initial states that are unseen during training, rNCA is not able to recover the ground truth as they require long-range reasoning. When the image is too ambiguous, local context may be insufficient to learn the correct dynamics. An example of this is shown in Supplementary Figure 7, where the initial predictions and corresponding image do not contain enough information for the rNCAto effectively improve the segmentations.”

---

### Official Review · Reviewer_VPHF · 2026-01-10

**Confidence:** 4
**Preliminary Rating:** 4
**Final Rating:** 5

**Summary:**

The authors present a method built on neural cellular automata (NCA) for refining segmentations, called rNCA. Unlike the existing methods where one must play with uninformed parameters such as adjusting threshold parameters for target prediction or morphological operations to improve segmentations, the iterative strategy of rNCA repairs topological information for a diverse range of segmentation (missed/improper) targets. The detailed results demonstrate how segmentation results can be improved and how rNCA does a good job at it.

**Strengths:**

1. I like how well-written and described the paper and corresponding experiments (and figures) are.
2. One of the most critical experiments demonstrating the strengths of rNCA is the comparison with UNet + UNet, used back to back to improve the pseudo labels.
3. I appreciate the simplicity of explaining the advantage of NCA in the manuscript and releasing the codebase for users to run the scripts, especially the application of rNCA over n-refinement-iterations (eg. Figure 4).

Overall, I have a very good impression about the manuscript.

**Weaknesses:**

There are a mainly two questions I'm interested to hear more about:

1. NCAs are known for their efficiency quotient. I am curious how efficient is it to train rNCA in three terms: a) in terms of training time, b) in terms of the amount of data, and c) in terms of compute resources.
2. Can rNCA be plugged to pretrained models? (eg. if a user has a pretrained nnUNet model, can one plug rNCA and refine their segmentations effortlessly?)

(see "Questions To Address In The Rebuttal" for more details outlined)

**Detailed Comments:**

Overall, I am confident that the paper is in a great state to be accepted at MIDL as it is, subject to some minor questions to be addressed and address them in the manuscript. I would love to see the response from the authors for the comments shared above. Thanks!

**Justification Of Final Rating:**

I am quite happy with the manuscript and the open-sourcing motivation. The authors did a great job at answering the overalls for all reviewers and engaging in discussion with me. Thank you so much for all the efforts put forward by the authors.

**Justification Of The Preliminary Rating:**

I am happy with the overall methods developement, presentation of the manuscript and the open-sourcing motivation of the entire codebase. I am open to discuss the response from authors in the coming days.

**Questions To Address In The Rebuttal:**

1. I found some parts in the manuscript talking about rNCA in terms of efficiency, but couldn't consolidate an overall opinion about it (again, solely in terms of efficiency). I suggest the authors add / update a section to enunciate a) how efficient rNCA is in terms of memory / parameter / amount of data / compute resources, and b) how all points in point b) compete with a UNet + UNet design. This would clearly demonstrate that rNCA scales beyond the manuscripts' presented qualitative and quantitative applications, i.e. imo the prime selling point of NCAs.
2. I didn't closely check the UNet implementation, but as per the explanation in Appendix; "The UNet is configured with five encoder levels with channel sizes (16, 32, 64, 128, 256) and downsampling strides of 2 at each level", it seems like the depth is down to 5 levels, assuming the spatial shapes of inputs are >= (256, 256). My question here would be, could the segmentation errors be due to such severe downsized pixel resolution?
3. The previous point leads me to my next question: how well does rNCA work with pretrained models? (eg. say pretrained nnUNet model on ACDC data). nnUNet is well known to work well for segmentation tasks, and is a standard framework for developing segmentation methods. My question here would be, how does nnUNet + nnUNet vs nnUNet + rNCA compare? I strongly suggest the authors to show results in this intersection to show how rNCA works in the wild.
4. I like the overall message from Figure 2., however it's a bit unclear to read whether the models (i.e. UNet + rNCA) are jointly trained or in two stages? (pointing to the previous Point 3) I suggest the authors clarify this either in the caption, or improve the figure to make this self explanatory.

I have one final (curious) question: what are the current limitations of rNCA? I spotted the volumetric scaling and multiple error handling discussions, however I am very curious about the failure cases (if at all), and what could be a plausible hypothesis in situations where rNCA indeed does fail! (only in absolute terms, not speaking relatively at all)

---

> ### Author Response · Authors · 2026-01-23
>
> We thank the Reviewer for their enthusiasm for this paper.
>
> # rNCA Efficiency
> In terms of training efficiency, we found that rNCA does not converge faster than conventional baselines, likely due to the iterative nature of this approach. Specifically, training times are longer than those of a UNet pipeline, e.g. time per epoch on DRIVE; UNet: 23.66s/epoch vs rNCA: 14.33s/epoch, with the rNCA taking twice as many epochs to reach convergence. While these results are anecdotal, we expect future work to more thoroughly investigate strategies for improving training convergence, which was not the main objective of this study.
>
> The primary efficiency advantage of rNCA lies at inference time and in model complexity. rNCA uses significantly fewer parameters and a smaller memory footprint than the other baselines, including U-Net. We have included a reference to inference runtimes and the number of parameters in Supplementary (Table 5). While inference involves an iterative rollout and is therefore marginally slower (2.48 ms vs 2.05 ms per image) than a single forward pass of a UNet* approach on GPU, the overall memory consumption is substantially lower, and the parameter count is reduced by an order of magnitude.
>
> To make this clear to the reader, we have updated both points in the discussion.
>
> # UNet Configuration:
> To ensure a strong baseline comparison to our experiment, we tried different U-Net configurations and this one yielded the best Dice on DRIVE. Hence we employed it for all our experiments. The UNet is implemented with skip connections, so it is able to use features from each downsampling layer, making downsampling not a limitation of this implementation.
>
> # rNCA on pretrained models
> The rNCA refinement network is designed to be trained entirely independently of the initial segmentation model (if one is used), meaning that it can be applied to refine the outputs of any state-of-the-art (SOTA) model. So yes, the rNCA can be used to refine any pre-trained model predictions.
>
> To address the experiment suggested by the reviewer, training nnUNet in the first stage and refining its predictions in a subsequent stage with either another nnUNet or rNCA, we note that this setup is conceptually analogous to the refinement experiments already reported in our study on the DRIVE dataset (e.g. the U-Net + U-Net*, Swin +U-Net* , and U-Net +rNCA experiments). In these experiments, rNCA demonstrated its model-agnostic behaviour by improving predictions generated from both a UNet and Swin, which can produce different error patterns. Further, we show that refining an initial model's predictions (e.g. U-Net or Swin) with the U-Net*, results in minor changes from the initial models performance (Table 1). We believe this is because the U-Net* relies on the initial prediction and gets stuck in a local minimum early in training, due to the complexity of the network. We would also expect to see a similar behaviour in a nnUNet+nnUnet* experiment. To make this clear, we have added this discussion to Section 4.2.
>
> While we acknowledge that nnUNet is widely adopted in medical imaging, introducing an additional comparison (e.g., nnUNet followed by nnUNet* versus nnUNet followed by rNCA) would largely replicate the existing UNet-based experiment and could unnecessarily increase the manuscript’s complexity. To prevent any potential confusion, we have updated the text to clearly state that rNCA is trained independently (see the next comment) and can effectively refine outputs from off-the-shelf pretrained segmentation models.
>
> # rNCA Training Procedure
> We thank the Reviewer for raising this point and agree that Figure 2 is currently ambiguous. To clarify, the refinement is trained completely independently. Initial masks are required for rNCA to refine, however, this process is done separately. We produced some initial masks by perturbing labels (Liver and ACDC) or using predictions from commonly used segmentation models (such as U-Net and Swin). To make this clear, we have updated Figure 2 to not show the initial segmentation and avoid this confusion. Now, Fig.2 only showcases the refinement process.
>
> Furthermore, we have updated the main text in Section 3 to explicitly mention the decoupled nature of the refinement process w.r.t the initial segmentation mask generation.
>
> # rNCA failure modes
> A key limitation of refinement methods is their reliance on the initial segmentation and the training distribution of mask errors. When initial states are severely broken and unseen during training, the model struggles to improve them. Hence, when test-time errors differ from training (Supplementary Figure 7), performance degrades.
>
> We believe another key limitation is the poor propagation of global information. While hidden states can transmit some global context, this capacity is limited. rNCA could benefit from having it’s local decision more globally informed, e.g. enabling cells to know the direction to the nearest anatomical structure.

---

> ### Comment · Reviewer_VPHF · 2026-01-30
>
> I thank the authors for taking their time to formulate the rebuttal and addressing feedback from all the reviewers.
>
> A few side notes from my side:
>
> > rNCA Efficiency
>
> The takeaways presented are quite interesting to me in general. The parameter count and inference runtime efficiency is great to hear indeed. For the training efficiency though:
>
> _"training times are longer than those of a UNet pipeline, ..., with the rNCA taking twice as many epochs to reach convergence. "_
>
> I'd be interested to hear your hypothesis for this (doesn't need to be quantified in any ways). Do you suspect some reasons in particular which could be the reason for this? (besides the iterative nature of the approach)
>
> > UNet Configuration:
>
> Thanks for the clarification. If the authors still have the results around for this, could you add them in a supplementary section in the camera-ready version? (no worries if the results aren't around. I just feel that the paper would profit with such a detailed experiment, as the paper enunciates a head-on comparison with UNet)
>
> > rNCA on pretrained models
>
> _"To prevent any potential confusion, we have updated the text to clearly state that rNCA is trained independently (see the next comment) and can effectively refine outputs from off-the-shelf pretrained segmentation models."_
>
> I agree it's a lot of experiments. My questions was more in the direction of how rNCA plugs in with a pretrained model off the shelf (no training done solely for the paper for the pretrained segmentation model). This would definitely make the paper fly to have a plug-and-play presentation, but I don't think it's a deal breaker, considering all the experiments presented by the authors.
>
> (hint: see my "Weakness" point: **Can rNCA be plugged to pretrained models? (eg. if a user has a pretrained nnUNet model, can one plug rNCA and refine their segmentations effortlessly?)** ;))
>
> > rNCA Training Procedure
>
> Thanks for the clarification.
>
> > rNCA failure modes
>
> I sincerely appreciate the takeaway presented here.
>
> _"We believe another key limitation is the poor propagation of global information. While hidden states can transmit some global context, this capacity is limited. rNCA could benefit from having it’s local decision more globally informed, e.g. enabling cells to know the direction to the nearest anatomical structure."_
>
> Ah interesting. This is good to know. Thanks for sharing your hypothesis about this.
>
> Once again, I thank the authors for the overall efforts put towards the manuscript. I am happy about the updates as is and would love to discuss about the two open notes here (in any case, I can understand the time pressure and acknowledge the efforts again).

---

> > ### Author Response · Authors · 2026-01-30
> >
> > We are really grateful for all the comments and suggestions of the reviewer, and very appreciate their interest on our work!
> > Here we try to answer his remaining comments to the best of our abilities:
> >
> > # rNCA Efficiency
> > We hypothesize that convergence is slower because the model is required not only to refine the initial masks, but also to learn to refine its own predictions over when they are reintroduced into the training pool. This feedback mechanism increases the diversity of the effective input distribution, as the set of possible initial states expands beyond the original data distribution. Hence, while the network architecture itself remains relatively simple, the training procedure induces additional complexity which we believe slows convergence. Related work from the Neural Cellular Automata (NCA) community, particularly efforts focused on improving training efficiency (Faldor, Maxence, and Antoine Cully. "Cax: Cellular automata accelerated in jax."; 2024), suggests that there may be principled strategies to mitigate these effects, which we view as a promising and exciting direction for future work.
> >
> > # UNet Configuration
> > Unfortunately we don’t have any bookkeeping of the prior UNet configurations, but we agree they would have been nice to show alongside the paper.
> >
> > # rNCA on pretrained models
> > We thank the reviewer for clarifying the requested experiment. We now see that the reviewer was asking about plugging rNCA into an off-the-shelf pretrained model instead of a model trained specifically for this paper. We apology for the confusion. Note however, that on the ACDC Dataset, we did not train our own models for the initial segmentation, instead we used published predictions of previous work (explained on the appendix).
> > “For the base segmentation models, we used the published predictions from transformer based Swin-UNETR (Cao et al., 2022) and U-Net (Ronneberger et al., 2015).”
> > Here, the rNCA was only trained on manually perturbed labels simulating the errors we want it to fix at test time. We will make this more clear in the main text for the camera ready submission.
> >
> > Admittedly we do not compare those refinements with other refinements as we did not see a conceptual difference to our experiments on DRIVE. We agree that it is a critical point to showcase the versatility of rNCA. An interesting future direction would be to create an off-the-shelf refiner by replicating the most common artifacts encountered in the wild through label perturbation alone!

---

> > > ### Comment · Reviewer_VPHF · 2026-01-30
> > >
> > > Thanks for your responses and engaging in the discussion!
> > >
> > > > rNCA Efficiency
> > >
> > > I see. Thanks for sharing your hypotheses. That's indeed great to hear!
> > >
> > > > UNet Configuration
> > >
> > > No worries at all.
> > >
> > > > rNCA on pretrained models
> > >
> > > _"Note however, that on the ACDC Dataset, we did not train our own models for the initial segmentation, instead we used published predictions of previous work (explained on the appendix). “For the base segmentation models, we used the published predictions from transformer based Swin-UNETR (Cao et al., 2022) and U-Net (Ronneberger et al., 2015).”"_
> > >
> > > Okay I missed it, apologies for that. And that's great (you've convinced me even more now ;))
> > >
> > > _"We will make this more clear in the main text for the camera ready submission."_
> > >
> > > That would be great. Thanks!
> > >
> > > _"An interesting future direction would be to create an off-the-shelf refiner by replicating the most common artifacts encountered in the wild through label perturbation alone!"_
> > >
> > > It's definitely gonna be a worthwhile effort. Something like "Refine Anything" ;)
> > >
> > > Thanks once again for taking your time to engage in discussion. I am even more confident about my updated ratings now!

---

### Author Rebuttal · Authors · 2026-01-23

**Rebuttal:**

We thank all reviewers for their careful reading of the manuscript and for the constructive and insightful feedback. We are also very pleased that all reviewers found the work novel and interesting, and we appreciate their recognition of the potential of rNCA-based refinement. We have addressed each comment in detail and revised the manuscript accordingly (changes in red). We believe the paper has substantially improved as a result of these revisions.

One of the main goals of the revision was to make the methodology more accessible, particularly for readers who are not experts in Neural Cellular Automata (NCA). To this end, we clarified the methodological descriptions, including a more detailed explanation of the inference, and updated Figure 2 accordingly. Similarly, a major revision concerns the clearer separation between the base segmentation stage and the rNCA-based refinement stage. We now explicitly emphasise the decoupled nature of these components (Fig. 2), clarifying that rNCA operates purely as a local refinement mechanism and doesn’t require end-to-end workflows. This distinction helps position the method more clearly with respect to existing end-to-end NCA and UNet-based approaches, and better communicates the intended role and scalability of rNCA. These additions aim to reduce ambiguity and improve readability without changing the technical scope of the work.

Finally, we have added and discussed failure cases of the proposed method (previously in the appendix). These highlights in which scenarios the refinement process may not yield improvements, and they help delineate the current limitations of rNCA.

Overall, these revisions have improved clarity, structure, and completeness, and they substantially increase the quality and accessibility of the manuscript. We hope that the revised version adequately addresses the reviewers’ concerns and look forward to further feedback.

**Supporting Material:**

/attachment/5dc63167b85b1e29c8977337c1f3b3872536442b.pdf

---

### Meta-Review · Area_Chair_kBWM · 2026-02-01

**Recommendation:** Accept (Poster)
**Confidence:** 4

**Metareview:**

This paper introduces rNCA, a neural cellular automata–based refinement method for repairing topological errors in segmentation masks through local, iterative updates guided by image context. The approach is conceptually novel in positioning NCA as a general, model-agnostic post-processing mechanism rather than an end-to-end segmentation model. Reviewers consistently found the idea original and well motivated, with strong experimental evidence across multiple datasets demonstrating improvements in Dice/clDice, topology-aware metrics (e.g., Betti errors), and boundary distances. The method is lightweight, parameter-efficient, and easy to integrate into existing pipelines, and the paper is clearly written with high-quality figures and open-sourced code.

The main limitations discussed by reviewers relate to reliance on the quality of the initial segmentation, sensitivity to hyperparameters, and the inherently local nature of the refinement dynamics, which can limit recovery in severely corrupted cases. These concerns are appropriately acknowledged and addressed by the authors through additional experiments, clearer methodological explanations, failure case analysis, and expanded discussion following the rebuttal. Overall, the paper makes a strong and original contribution to segmentation refinement, is technically sound, and is well aligned with the scope and interests of MIDL.

---

### Decision · Program_Chairs · 2026-02-13

Accept (Poster)